# Using set visualisation to find and explain patterns of missing values: a case study with NHS hospital episode statistics data

Roy A Ruddle [ORCID],[1] Muhammad Adnan [ORCID],[2] Marlous Hall[3]

[1]School of Computing and Leeds Institute for Data Analytics, University of Leeds, Leeds, UK
[2]Computer Science, Higher Colleges of Technology, Sharjah, UAE
[3]Leeds Institute of Cardiovascular & Metabolic Medicine and Leeds Institute for Data Analytics, University of Leeds, Leeds, UK

**Correspondence to**
Professor Roy A Ruddle;
r.a.ruddle@leeds.ac.uk

## ABSTRACT

**Objectives** Missing data is the most common data quality issue in electronic health records (EHRs). Missing data checks implemented in common analytical software are typically limited to counting the number of missing values in individual fields, but researchers and organisations also need to understand multifield missing data patterns to better inform advanced missing data strategies for which counts or numerical summaries are poorly suited. This study shows how set-based visualisation enables multifield missing data patterns to be discovered and investigated.

**Design** Development and evaluation of interactive set visualisation techniques to find patterns of missing data and generate actionable insights. The visualisations comprised easily interpretable bar charts for sets, heatmaps for set intersections and histograms for distributions of both sets and intersections.

**Setting and participants** Anonymised admitted patient care health records for National Health Service (NHS) hospitals and independent sector providers in England. The visualisation and data mining software was run over 16 million records and 86 fields in the dataset.

**Results** The dataset contained 960 million missing values. Set visualisation bar charts showed how those values were distributed across the fields, including several fields that, unexpectedly, were not complete. Set intersection heatmaps revealed unexpected gaps in diagnosis, operation and date fields because diagnosis and operation fields were not filled up sequentially and some operations did not have corresponding dates. Information gain ratio and entropy calculations allowed us to identify the origin of each unexpected pattern, in terms of the values of other fields.

**Conclusions** Our findings show how set visualisation reveals important insights about multifield missing data patterns in large EHR datasets. The study revealed both rare and widespread data quality issues that were previously unknown, and allowed a particular part of a specific hospital to be pinpointed as the origin of rare issues that NHS Digital did not know exist.

## INTRODUCTION

Missing data occurs more often in electronic health records (EHRs) than any other single data quality issue.[1] The motivating problem for the present research was how to investigate patterns of missing data that involve many variables and explain those patterns by identifying the underlying structures.[2] Our objective was to develop and evaluate a method to achieve that using set visualisation, to enable users to generate actionable insights from datasets that contain millions of records. To be successful, the method needs to reveal both known and unknown patterns, including those that are often obscured in a dataset due to their rarity. To help ensure that the visualisation method was scalable we chose a set-based approach, but limited the scope of the research to analysing flat file data tables. The research used the transparent reporting of a multivariable prediction model for individual prognosis or diagnosis reporting guidelines.[3]

Researchers need to investigate patterns of missing values to understand possible effects on cohort selection, bias, impact on data linkages and to design appropriate missing data techniques such as multiple imputation by chained equations.[1 4–8] Organisations collecting and providing such data strive to continuously improve data quality.[9–12] We collaborate with one such organisation, NHS Digital, which is the UKs national information

### STRENGTHS AND LIMITATIONS OF THIS STUDY

⇒ This study demonstrates the utility of interactive set visualisation techniques for finding and explaining patterns of missing values, irrespective of whether those patterns are common or rare.
⇒ Techniques were evaluated with a large national admitted patient care dataset (16 million records across 86 fields).
⇒ Evaluation only involved one single-table dataset, but that was from a national organisation providing many similar data extracts each year to researchers and organisations.
⇒ Evaluation of the interactive set visualisation techniques did not involve formal usability testing.

and technology service collecting more than 100 million records/year of Hospital Episode Statistics (HES)[13] for the National Health Service (NHS) in England. HES data are directly used in the payment of hospitals throughout England, as well as for secondary uses such as research, quality indicators and planning health services.[8 14]

At present, the only checks that organisations typically perform are basic counts of missing values in individual fields,[6 7 9 12 15–17] with researchers then implementing a range of techniques from removing records with missing values, dropping variables or more advanced analytical techniques including multiple imputation by chained equations and bootstrapping integrated into their analyses.[4 7 15] In NHS Digital's case, those checks are limited to core data fields (eg, NHS Number), and if the percentage missing exceeds a threshold (30%) then feedback is sent to the hospital.[10 12] However, researchers also want to investigate patterns of missing values that involve multiple fields,[1 18] and NHS Digital wants to do that to define new business rules for data cleaning and to cross-reference variations in data quality with external influences (eg, changes in policy or coding improvement initiatives).

Patterns of missing values may be computed and/or visualised.[2] Some computations model the distribution of missing values to estimate whether they adhere to a predefined pattern and to allow the analysts to consider assumptions about missing at random, missing completely at random or missing not at random patterns,[19] with the latter including monotone and unit non-response ('block') patterns.[20] However, these methods rely on probabilistic models, so by their very nature, they are poorly suited for detecting rare patterns of missingness.

An alternative approach is to use set analysis. Imagine a dataset has three fields (A, B and C). Set A is all of the records that are missing the value for field A, and similarly for sets B and C. There are four possible exclusive set intersections (AB, AC, BC and ABC), where intersection AB contains the records that are only missing values for fields A and B, and so on. The number of intersections cannot be greater than the number of records in the dataset,[21] so it is computationally tractable to identify every combination of fields that are missing together irrespective of whether it occurs millions of time or only once.

Sophisticated visualisation tools and techniques have been developed for cohort selection, risk analysis and other types of detailed EHR data analysis,[22–24] but data quality use cases are a notable omission.[25] Visualisation dashboards are widely used in healthcare,[26] but focus on high-level metrics such as the number of values that are missing from key fields, which corresponds to visualising specific sets of missing data. However, it is not sufficient for tools to visualise sets of missing data because users need to visualise set intersections if they are to investigate multifield patterns of missing values in detail. Most tools to date adopt designs that are unsuitable for showing intersections that involve many sets because Venn diagram or node-link representations are used,[27–29] only display pairwise intersections,[30 31] or require excessive interaction because users have to select specific fields of interest.[32–34]

The most appropriate technique for visualising set intersections is a matrix of sets versus intersections. That technique is provided by VIM,[35] but only as static plots. UpSet[36] provides interactive set intersection visualisation but, when the present research was conducted, the latest version (UpSet2) only allowed 5 MB of data to be loaded and our dataset was 1000 times larger. Since then, more scalable R and Python UpSet implementations have been released[37 38] but, like VIM they also generate static plots.

## MATERIALS AND METHODS

### Setting and dataset

We used a 15 733 889 record extract of admitted patient care (APC) data from 2016 to 2017 HES from NHS Digital. APC data contain information about admissions to NHS hospitals in England and to independent sector providers paid for by the NHS, and is collected for clinical uses, determining payments to hospitals, care quality reports and a wide variety of research.[16] Our extract was from a project that was studying long-term outcomes and hospitalisation rates for acute myocardial infarction (heart attack) patients, and requested help investigating the quality of their data. The cohort comprised patients who were aged 18 or older at 31 March 2009, and had not had an acute myocardial infarction during years 2001/2002 to 2007/2008.

The dataset was processed to flag any 'unknown' values in each field[12] as missing, and had 86 fields that may be divided into four groups. Twenty fields contained diagnosis codes for a patient's illness or condition (DIAG_01 to DIAG_20), which should be filled in incrementally and exhibit a monotone pattern of missing data. Similarly, 24 fields contained codes about a patient's operations (OPERTN_01 to OPERTN_24) and 24 fields contained the corresponding dates for those operations (MYOPDATE_01 to MYOPDATE_24), with the operations and dates fields exhibiting monotone patterns, and each pair of operation/date fields exhibiting a block missingness pattern. There were 18 general fields: patient, episode and A&E record identifiers (ENCRYPTED_HESID; EPIKEY; AEKEY), episode order, status and type (EPIORDER; EPISTAT; EPITYPE), flags to indicate a finished admission and consultant episode (FAE; FCE), dates of admission, episode start and episode end (MYADMIDATE; MYEPISTART; MYEPIEND), patient date of birth, age and sex (MYDOB; ADMIAGE; SEX), whether patient was alive or dead and time from episode end date to death (Mortality; SURVIVALTIME), admission method (ADMIMETH) and the hospital's code (PROCODE).

### Visualisation design

We designed a novel tool called analysis of combinations of events (ACE), which is freely available for download and use.[39] The tool reads data from a flat file format (a CSV or other text file), calculates the exclusive set

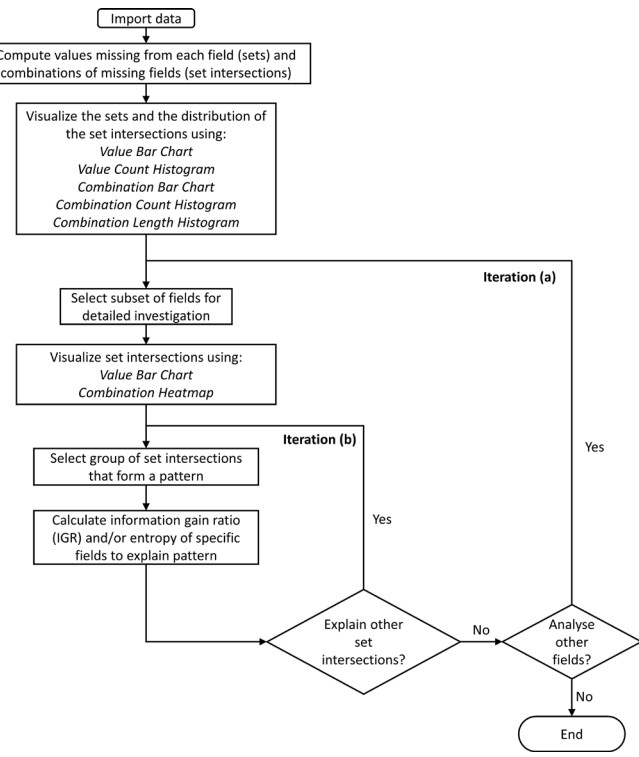

**Figure 1** A workflow for investigating the patterns of missing values with ACE (Analysis of Combinations of Events). The two levels of iteration are to: (a) select subsets of fields, (b) select groups of set intersections to explain a pattern in terms of the values of other fields.

intersections and stores the results in a memory-resident HyperSQL database.[40] ACE is implemented in Java and R, with a JavaFX graphical user interface.

ACE combines set-based visualisation and data mining methods in an integrated, interactive tool that enables users to investigate multifield missing data patterns.

Figure 1 shows a recommended workflow, which starts with computing the sets (one for each field) and set intersections (each combination of missing fields), and then visualising the sets (the number of values missing from each field) and distributions of the set intersections (to investigate high-level patterns). The substance of the workflow follows, with two levels of iteration.

The user interface is based on a tab pane layout that allows users to: (1) see each iterative step taken during the analysis, (2) undo steps by closing tabs and (3) compare patterns of missing values in different subsets of the data by switching tabs. An alternative to tabs would have been to integrate all of the visualisations into a single window, but that would have reduced the screen space available for each visualisation and limit the detail that users could see without scrolling or zooming.

Two of the visualisations are for sets and four visualisations are for set intersections. The primary set visualisation is called a value bar chart (figure 2), which shows the number of missing values in each field (this is standard functionality in other tools).[30–37 41–43] That visualisation is complemented by a value count histogram, which calculates bins for the number of missing values, shows the number of fields in each bin and lets our tool scale to datasets with an unlimited number of fields. Both of the visualisations allow users to sort the fields, and to select certain fields for the next step in the analysis.

The primary set intersection visualisation is called a combination heatmap (figure 3), which shows fields (X axis)×set intersections (Y axis) and the number of records that are in each intersection (colour). The height of the heatmap depends on the number of intersections, which makes it suitable for large datasets. By contrast, a missingness map[44] displays a separate row for each record in a dataset, so the case study's 16 million record dataset would

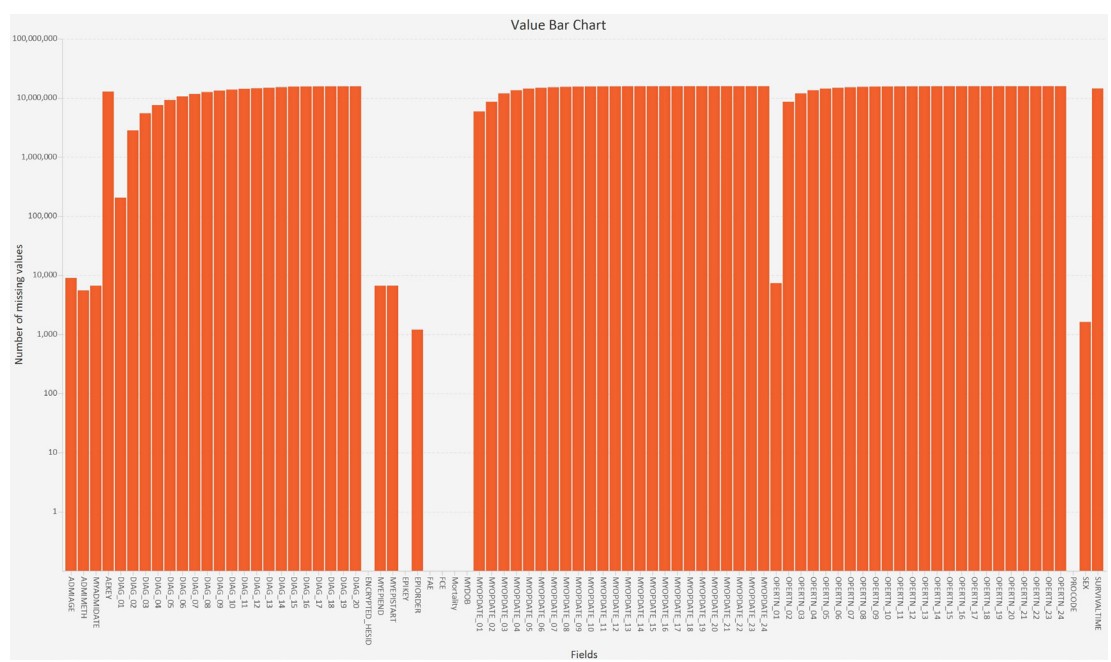

**Figure 2** A value bar chart showing the sets (the number of missing values in each field; NB: the Y-axis uses a log scale).

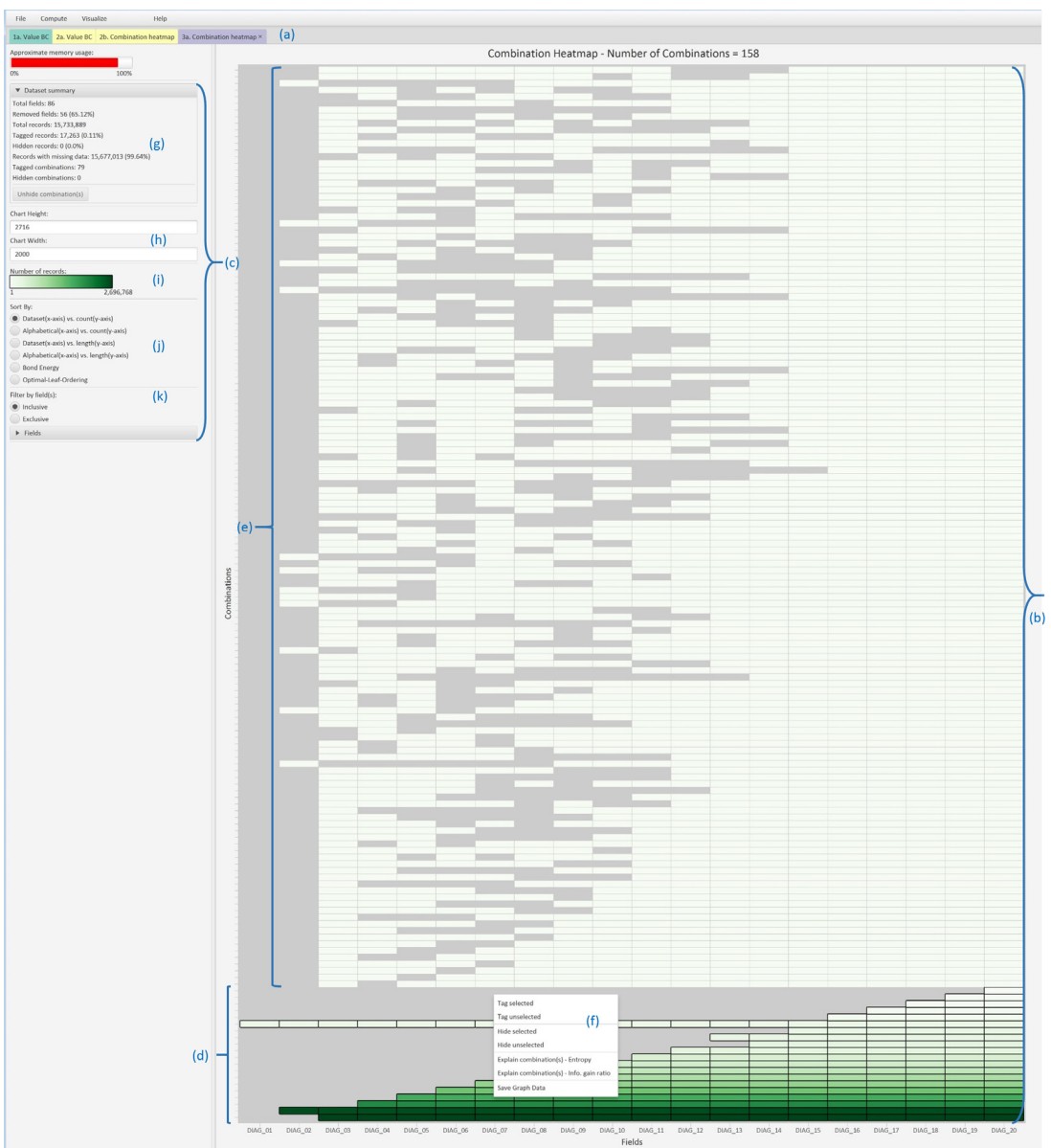

**Figure 3** ACE (Analysis of Combinations of Events) uses a tab pane layout with color-coded tabs (a). Each tab is split into a visualisation panel (b) and a control panel (c), and the details depend on the type of visualisation. This figure shows a heatmap of expected (d) and unexpected (e) set intersections that involve the DIAG_nn fields. A context menu (f) highlights various actions that can be performed on a selected intersection. The control panel contains a dataset summary (g), visualisation size (h), heatmap legend (i), sorting methods (j) and filtering options (k).

produce a map with 16 million rows, which is clearly not practical.

The heatmap fields may be sorted alphabetically or into dataset order, the intersections may be sorted by degree (number of missing fields) and cardinality (number of records) and the heatmap cells may be clustered using two heuristic sorting methods from a recent state-of-the-art report.[45] The heatmap is complemented by a combination count histogram and a combination length histogram (figure 4) that calculate bins for the number of records and fields, respectively, and show the number of intersections in each bin. This lets ACE scale to an unlimited number of set intersections. The sixth visualisation (a combination bar chart) is for intermediate

scalability because it shows the number of records in each intersection.

The tool's data mining methods comprise information gain ratio (IGR)[46] and entropy calculations, which are key concepts in information theory,[47] IGR measures the change in entropy when a field is used to classify data, which enables users to rank fields in terms of their ability at explaining certain intersections. Entropy quantifies how cleanly data are divided into different classes, which enables users to identify which values of a specific field are strongly associated with the intersections. IGR output was displayed in a bar chart in its own tab, and entropy output was displayed in a bar chart or table in its own tab.

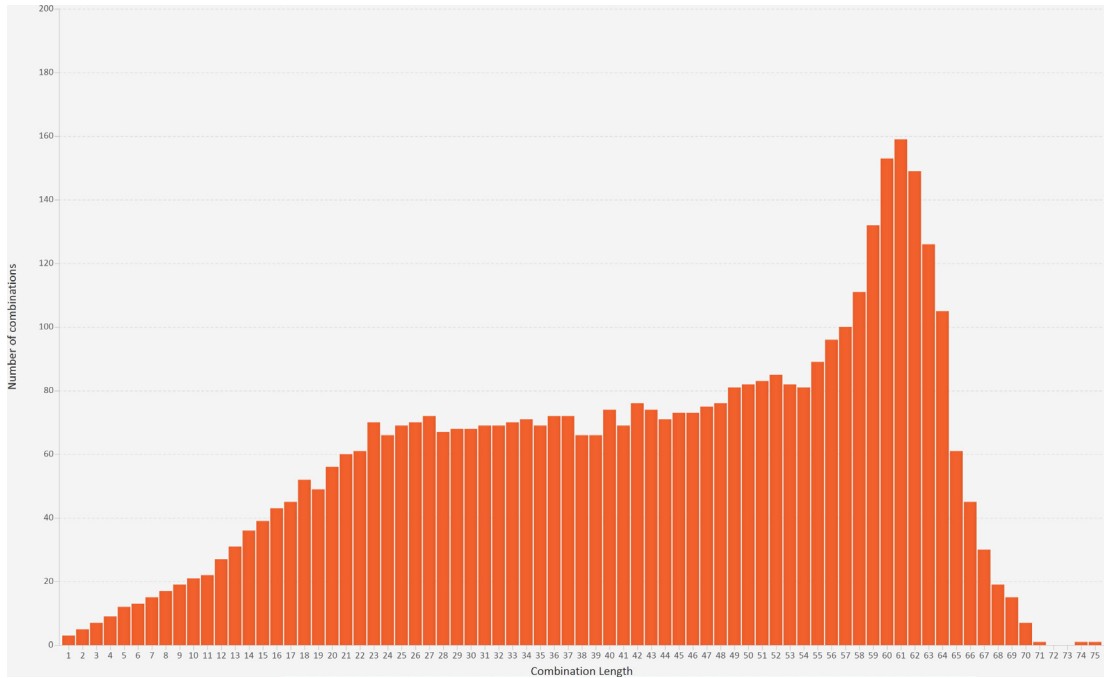

**Figure 4** A combination length histogram, which shows that most of the set intersections involved 23–64 fields.

All of the above visualisations and data mining techniques can be applied to either the whole dataset or subsets of it, chosen by selecting or filtering-out fields and/or combinations. This lets users iteratively analyse multiple patterns of missing values that coexist in the dataset.

### Case study

The ACE software was evaluated in a case study with a senior epidemiologist who was studying long-term outcomes and hospitalisation rates for survivors of acute myocardial infarction. During the case study, the epidemiologist and one of the software's developers worked together, with the latter controlling the user interface. This mimics the collaborative working environment that is found in many data science projects, with one person contributing domain knowledge and insights, while the other provides expertise in certain analysis methods and tools. Since then we have created training materials[39] and carried out training workshops to confirm that those materials and the software are easy for people to use independently. After the collaborative session with the epidemiologist, we discussed some of the insights with representatives from NHS Digital.

### Patient and public involvement

Neither patients nor the public were involved in the design, or conduct, or reporting, or dissemination plans of this research.

### RESULTS

The results are presented in a form that follows the workflow (figure 1). First, the dataset was loaded and the set intersections were computed. There were 4371

intersections, a total of 960 million missing values, and all except 120 records were missing at least one value.

A value bar chart showed that, as expected, there were three monotone patterns of missing data—one for the diagnosis codes, one for the operation codes and one for the operation dates (figure 2). However, the 4371 set intersections were many more than would occur if just those three monotone patterns were present, so we used combination count and length histograms to visualise the degree (number of missing fields) and cardinality (number of records) of the set intersections, respectively. This showed that 81% of the intersections involved 23–64 fields (figure 4), and 83% of the intersections were rare (each involving <0.1% of the records). Therefore, we did five iterations in the next stage of analysis (figure 1 iteration (a)) to investigate the patterns of missing values in the general fields, diagnosis codes, operation codes, operation dates and then operation codes and dates together. For each of those we performed one or more iterations to explain the patterns (figure 1 iteration (b)).

### General fields

For the general fields, a combination heatmap showed that the most common pattern was the 11 832 262 records that were just missing the SURVIVALTIME and AEKEY fields. The AEKEY links APC data with hospital accident and emergency (A&E) data, so the pattern is consistent with the fact that most patients survive and do not enter hospital through A&E. However, on seeing the pattern the epidemiologist commented that the survival time should not be missing at all because the data provider had been asked to derive it for all patients based on the study census date, regardless of their mortality status. Clearly that had not been done, so she would have to estimate

survival time aggregated to the nearest month in her survival analyses for those patients who did not die prior to the census date—thus losing precision of information compared with patients who had died prior to census. As well as loss of precision, this precluded analyses of short-term survival patterns within 30 days of admission.

From the value bar chart, the epidemiologist was also surprised to see EPIORDER and three date fields (MYAD-MIDATE, MYEPISTART and MYEPIEND) were some-times missing instead of being complete (figure 2). After selecting the two intersections that involved all three date fields from the heatmap, entropy calculations showed that all those records concerned maternity episodes (EPITYPE=5 'other delivery event' or EPITYPE=6 'birth event') and were the only records in the dataset with EPIORDER=98 ('not applicable'). EPIORDER was missing in three intersections (1210 records), and IGR calculations showed that the FAE and PROCODE fields were most likely to provide an explanation. Entropy calculations showed that EPIORDER was only missing for 10 healthcare providers and, for them, only when FAE=0 ('unfinished admission episode'). Unfinished episodes are common (3 million in the dataset), but EPIORDER was always provided by the other 553 healthcare providers.

### Diagnosis codes

The diagnosis fields should have just exhibited a mono-tone pattern, with 20 intersections showing that the codes were missing progressively more often from DIAG_01 to DIAG_20. However, a combination heatmap showed a dramatically different picture, with a total of 158 intersections and all of the additional ones contained gaps in the diagnostic fields (figure 3).

After interactively selecting the 138 unexpected intersections with gaps, another value bar chart showed that DIAG_03 was missing substantially more often than the subsequent diagnosis fields (figure 5). Entropy calcula-tions showed that all 1776 records in those intersections were for EPITYPE=3 ('birth episode'), all 1776 records were for ADMIMETH=31 ('admitted antepartum') and 1575 of the records were from one PROCODE.

### Operation code and date fields

We first investigated the operation codes and dates sepa-rately, and then together. The code fields exhibited the same characteristics as the diagnosis codes. A combina-tion heatmap showed that most (15 717 333) records exhibited a monotone pattern. However, there were also some gaps (72 set intersections; 2488 records). After we selected those intersections, IGR and entropy calcula-tions showed that 1795 of the records belonged to ADMI-METH=31 and 1588 of the records were from the same PROCODE as the diagnosis gaps.

The operation date fields also exhibited the same char-acteristics. Most (15 716 187) records exhibited a mono-tone pattern, but gaps occurred in 89 intersections (3689 records). Again, entropy calculations showed that 2458 of the gaps occurred for ADMIMETH=31 and 2159 of the records were from same PROCODE as above.

Further unexpected patterns were revealed when the operation codes and dates were visualised together, because 45 intersections had no gaps but involved a different number of code and date fields. Each operation code should have a corresponding date in every record (ie, a block missingness pattern), but 39 out of those 45 intersections contained fewer operation dates than codes, and the other six intersections contained extra dates.

The gaps in the diagnosis codes, and operation codes and dates fields were a surprise to the epidemiologist and to members of NHS Digital's Data Quality team, with whom we subsequently discussed the findings. Of particular concern to NHS Digital were the oper-ation codes, because they affect the process that the NHS Digital Casemix Team use to calculate healthcare resource groups (HRGs; the NHS equivalent of the diag-nosis related groups that were pioneered in the USA). HRGs group together clinically similar treatments that involve similar levels of healthcare resource and, therefore, are one of the building blocks of the NHS's Payment by Results system for hospitals.[48] That system is used because it is much simpler than determining the amount to be paid for each of the tens of thousands of

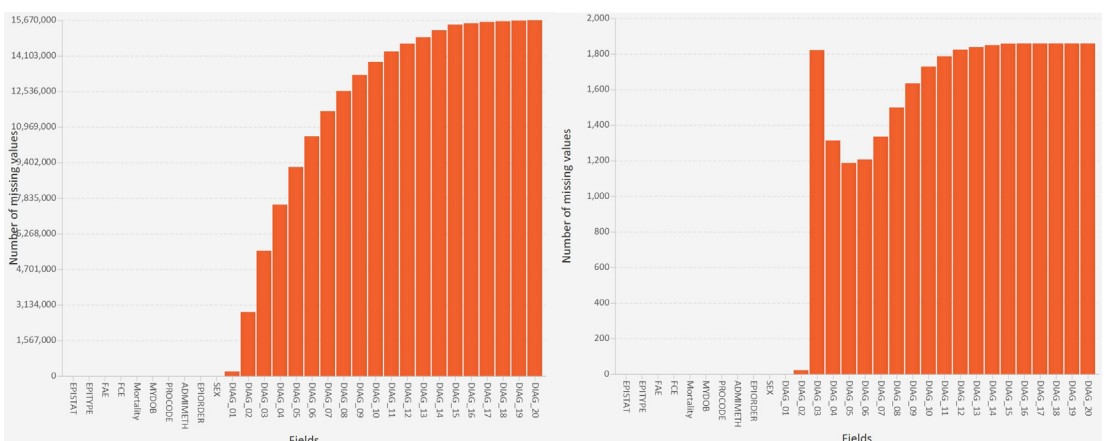

**Figure 5** A value bar chart showing the number of records in each set for the whole dataset (left) and the set intersections that, unexpectedly, contained gaps in the DIAG_nn fields (right).

interventions and diagnoses that are used for patients in hospitals.

## DISCUSSION

This study shows how set visualisation techniques may be used to investigate and explain patterns of missing data in tabular data. Set visualisation was applied by treating all of the records that were missing a given field as a set, so a set intersection comprised the records that were missing the same combination of fields. The two main visualisation techniques were bar charts to show each set (ie, the number of records missing each field) and heatmaps to show set intersections (ie, the number of records that were only missing each specific combination fields but did have values for the other fields). They were combined with histograms and another bar chart to let the tool scale to the millions of records and thousands of set intersections in the data. The visualisations were essential for enabling users to see and reason about the completely unexpected patterns, and integrating the visualisations with two well-known data mining methods (IGR and entropy) enabled users to pinpoint some of the patterns' origins and explain the underlying structures. The interactive tool we developed is freely available.[39]

Bar charts, heatmaps and histograms are standard visualisation techniques, of course. However, in the present study, it was the way those techniques and the data mining methods were combined into an easy-to-use interactive tool that was novel, and key to being able to apply set visualisation to find and explain the wide variety of patterns of missing data that occurred. To the authors' knowledge, the study is the first time that set visualisation has been applied to large-scale EHR data and the study's strengths are: (a) showing the benefits of different set visualisation techniques and how they combine to allow a holistic investigation of missing data patterns and (b) the actionable insights that those visualisations produced for both researchers (the epidemiologist) and organisations (NHS Digital and individual hospitals).

In terms of missing data, most researchers and organisations only count the number of values that are missing in individual fields,[6 7 9 12 15–17] showing the results in either a table or the equivalent of our value bar chart. In our study that bar chart showed the high-level patterns that were dominant (the well-known monotone patterns in the diagnosis codes, operation codes and operation dates), but also and contrary to expectations that a few fields were missing some values instead of being complete.

The most important visualisation was the combination heatmap, which we used to visualise set intersections. By using a heatmap, the unexpectedly large number of intersections and the multitude of gaps in the diagnosis codes, operation codes and operation dates all popped out (ie, were immediately obvious) to users, and closer inspection revealed the discrepancies between pairs of operation codes and dates. The tool's interactive data mining then enabled us to pinpoint the origin of some of the

unexpected patterns. The few pieces of data quality software that can visualise set intersections are either interactive but only handle small datasets,[36] or generate static plots and necessitate users writing extra computer code to select specific set intersections (eg, all those with gaps in the operation codes) for further investigation,[35 37 38] rather than interactively selecting those intersections with a few mouse clicks as with our ACE tool. Unlike ACE, none of that other software has any inbuilt data mining functionality.

The combination count and length histograms were useful during the first stage of analysis (figure 1), to help a user gain a general understanding of the interwoven nature of the patterns of missing data. Had the combination heatmap been used at that stage then the user would have been overwhelmed by the 4371 rows. Our dataset only had 86 fields, but had there been more then the value count histogram would have been similarly useful during the first stage of analysis.

The study had two main limitations. First, the case study only involved one dataset, although one from a national organisation that provides similar datasets to many researchers and organisations each year. Second, the tool is primarily designed to work with single data tables. However, longitudinal data would be easy to accommodate by concatenating each dataset and adding a field that stored a date for each dataset. Missing data patterns that occurred throughout the data would appear as patterns did in the present study, and patterns that were limited to a particular part(s) of the longitudinal data would be revealed by entropy calculations that involved the new field. Missing data patterns in a relational database could be investigated by either analysing each data table separately or joining fields of interest into a new table for analysis. For example, imagine a researcher's database contained information about general practitioner (GP) visits, diagnoses and prescriptions. The researcher could process the data to create a new table that contained fields about the visit (Date; Patient ID; GP ID), diagnoses and prescriptions (Drug; Dose). ACE's set intersection heatmap would show that as expected some visits resulted in a prescription whereas others did not, and unexpected patterns such as visits without a date or prescriptions without a dose would pop out as additional rows in the heatmap. The researcher could then use ACE's interactive data mining to determine whether the unexpected patterns were correlated with particular patients, GP practices or diagnoses.

Our approach benefits analysts and researchers by making it easier to check data when it is received. In principle that would have allowed the epidemiologist to remedy the survival time issue by a data request amendment, but the time limit for doing that (1 month) had passed before the case study started. Regarding the missing dates, as the tool pinpointed the origin as maternity records the epidemiologist made an informed decision to discard the records because she was conducting cardiovascular research. Our approach also benefits data

providers by pinpointing where issues occur (eg, the gaps originated almost entirely in a particular part of a specific hospital). Knowing that, it would be straightforward to provide feedback to rectify the problem via NHS Digital's existing data quality lifecycle[12] and appropriate actions (eg, fixing information technology (IT) issues, modifying processes or staff training). Our approach helped communication within NHS Digital, allowing a connection to be made between gaps in the operation code fields and the NHS's Payment by Results system[48] at a national level.

The present study showed how set visualisation enables users to identify a variety of unexpected missing data patterns. With further research, that could be developed into a comprehensive library of patterns that users can apply in health data settings to analyse missing data faster and more comprehensively. Further research is also needed to determine the data mining methods that are best suited for determining the structures that explain each type of pattern, extending our use of IGR/entropy and the work of others with decision trees.[2]

## CONCLUSIONS

Missing data has widespread impacts on secondary uses of EHRs. Investigations are typically limited to counting the number of values that are missing from key fields, and there is a lack of methods for investigating more complex patterns of missing values, and particularly patterns that are uncommon. A solution is offered by set visualisation techniques, which treat records that are missing a given field as a set, and records that are missing multiple fields as set intersections. This study describes how interactive set visualisation can be combined with data mining to identify and explain complex patterns of missing values. Using a case study with a 16 million record/86 field EHR dataset, we demonstrate that set visualisation revealed actionable insights that ranged from determining the data were not suitable for analysing short-term survival patterns, to providing feedback to hospitals to improve future data quality and knock-on consequences for hospital payments.

**Acknowledgements** The authors thank the NHS Digital staff who provided guidance during the research and comments about the implications of the findings.

**Contributors** MA and RAR designed the software. MA developed the software. MH provided subject matter expertise about HES data. RAR wrote the first full version of the manuscript. RAR is the guarantor of the manuscript and accepts full responsibility for the work and conduct of the study. All authors contributed to the case study and approved the final manuscript.

**Funding** This research was supported by the Engineering and Physical Sciences Research Council grant numbers EP/N013980/1 and EP/K503836/1, the British Heart Foundation grant number PG/13/81/30474 and the Alan Turing Institute.

**Competing interests** None declared.

**Patient and public involvement** Patients and/or the public were not involved in the design, or conduct, or reporting, or dissemination plans of this research.

**Patient consent for publication** Not applicable.

**Provenance and peer review** Not commissioned; externally peer reviewed.

**Data availability statement** Data may be obtained from a third party and are not publicly available. The dataset was provided by NHS Digital (request number DARS-NIC-17649-G0X4B-v0.6) and, due to data governance restrictions, cannot be made openly available.

**ORCID iDs**
Roy A Ruddle http://orcid.org/0000-0001-8662-8103
Muhammad Adnan http://orcid.org/0000-0001-7706-6604

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
