## [Reviewer comments · BMJ Open]

ARTICLE DETAILS

TITLE (PROVISIONAL)	Using set visualization to find and explain patterns of missing values: A case study with NHS hospital episode statistics data
AUTHORS	Ruddle, Roy; Adnan, Muhammad; Hall, Marlous

VERSION 1 – REVIEW

REVIEWER	Cornish, Rosie University of Bristol, School of Social and Community Medicine
REVIEW RETURNED	12-Jul-2022

GENERAL COMMENTS	This manuscript describes a tool that can be used to investigate patterns of missing data in electronic health records or other large datasets and demonstrates its use using Hospital Episode Statistics data. The tool is described very clearly and I agree that it could prove extremely useful in investigating patterns of missing data in such datasets. I do feel that some specific comments in the paper need to be qualified slightly – please see below. Main comments 1. In the introduction (page 6, lines 3-4), discussion (page 14, lines 22-23) and conclusions (p. 15, lines 10-11) the authors state that “researchers and organisations” typically only count the number of missing values in individual variables. The references they give are either about assessing the quality of administrative or routine health data in general or about assessing missingness in a specific variable (reference 6). There are many researchers who use electronic health data for applied health research and a certain proportion of these (but by no means all) are likely to investigate missing data in much more detail than this, use complex statistical methods to address missing data, and so on. I therefore feel that this statement needs qualifying somehow. Do the authors mean researchers whose specific aim is to assess quality and/or quantify missing data in such datasets, or all researchers using these datasets for any purpose? If the latter, I feel that they need to include some references to support this.2. Different datasets containing electronic health records are quite different in the way they are structured/collected. For example, in the HES data there are many variables that should be non-missing (e.g. admission date etc) and, as indicated in this manuscript, the diagnosis and other codes should follow a predictable pattern of missing data. However, in other electronic health records (for example, UK data from general practices such as those held by the Clinical Practice Research Datalink) every GP encounter should be date stamped but most of the data recorded will relate to whatever information was recorded during that encounter (diagnoses, symptoms, treatments, tests, and so on). Some data items should result in others being recorded (e.g. a prescription
---

	should be accompanied by information about dose etc), in which case it would be possible to examine missing data in the way described; however, it is more difficult to see how useful the set visualisation tool would be for such datasets. I feel that this should be acknowledged somewhere in the manuscript – i.e. that although the tool may be extremely useful for some datasets, it may be less well suited to others as this depends very much on how the data are structured and collected. Minor comments  1. As I understand it, this manuscript is partially aimed at epidemiologists / applied researchers using electronic health data for their research. Although entropy and information gain ratio are key concepts in information theory, they may not be widely known to epidemiological researchers. These terms are explained in this particular context but I wonder if it is worth giving a more general definition in the paper? 2. Some acronyms are used but not defined (e.g. ESI)
--	---

REVIEWER	Wu, Danny University of Cincinnati, Biomedical Informatics
REVIEW RETURNED	01-Sep-2022

GENERAL COMMENTS	The purpose of this paper was to demonstrate the effectiveness of ACE, a novel R and Java based interactive visualization tool, at identifying missing data from large scale electronic health records (EHRs) in the UK's NHS digital. This tool included bar chart and heatmap visualization sets with data mining techniques to provide researchers with insights in missing patterns. The results showed that ACE is effective in identifying widespread and rare data quality issues. The study can be utilized for both epidemiologists and hospital services to improve data quality and downstream analyses. Overall, the paper is well written with the purpose and the relevance clearly identified. The implications and limitations of the present study are discussed in detail. While the case study and methodology need more specificity, the authors created comprehensive and relevant figures to allow the readers to have a greater visual understanding of the study design. My suggestions are listed below. P4: A brief description of the methodology (e.g., visualization design choices and data mining techniques). is not given in the abstract. P4 L3-5: This sentence could be better worded for clarity. Is there supposed to be a "that" after "understand"? P4 L9: Are the 16 million records freely available to UK researchers or any researchers? P4 L14: Give an example of "unexpected gaps" in the study. P4 L19: When stated "previously unknown to an Epidemiologist" should be stated as "previously unknown in the Epidemiology field". Otherwise, it implies that one epidemiologist instigated the study and does not emphasize its relevance. P5 L8: Would the ACE be applicable to a different dataset? If so, what data manipulation effort would be needed? P5 L22: Spell out TRIPOD when it is first used. P6 L4: What are the "core data fields" that basic counts only address?
--

	P6 L29: Provide examples of the existing tools to explain why their information and designs are not suitable to illustrate set intersections. P7 L22: When examining the data set, missing data may come from a smaller set of hospital practices. In other words, hospitals may have separate internal issues in data collection processes, leading to different prevalence of missing data. Consider addressing this in the discussion. P9 L21: Explain at high level the information gain ratio and entropy calculations. P14 L13: Consider adding another limitation in human-centered evaluation or formal usability testing, where multiple epidemiologists and other stakeholders are invited to perform tasks using the ACE to identify missing data patterns and provide feedback to the usability of the tool.
--	--

VERSION 1 – AUTHOR RESPONSE

Reviewer: 1

Dr. Rosie Cornish, University of Bristol Comments to the Author:

This manuscript describes a tool that can be used to investigate patterns of missing data in electronic health records or other large datasets and demonstrates its use using Hospital Episode Statistics data. The tool is described very clearly and I agree that it could prove extremely useful in investigating patterns of missing data in such datasets. I do feel that some specific comments in the paper need to be qualified slightly – please see below.

Main comments

1. In the introduction (page 6, lines 3-4), discussion (page 14, lines 22-23) and conclusions (p. 15, lines 10-11) the authors state that “researchers and organisations” typically only count the number of missing values in individual variables. The references they give are either about assessing the quality of administrative or routine health data in general or about assessing missingness in a specific variable (reference 6). There are many researchers who use electronic health data for applied health research and a certain proportion of these (but by no means all) are likely to investigate missing data in much more detail than this, use complex statistical methods to address missing data, and so on. I therefore feel that this statement needs qualifying somehow. Do the authors mean researchers whose specific aim is to assess quality and/or quantify missing data in such datasets, or all researchers using these datasets for any purpose? If the latter, I feel that they need to include some references to support this.

Many thanks for highlighting this, and we appreciate our original wording did not capture the full sentiment. Many researchers unfortunately don’t use advanced or complex missing data techniques, but there are experienced researchers who do. The main point we are trying to convey is that whether or not researchers are experienced in advanced missing data strategies – the standard and available software doesn’t often have easily implementable tools to look at the complex patterns of missingness across multiple fields, in particular for in very large EHR data – and it is helpful and important to have the ability to look at more detailed summaries of missing data to better inform such advanced techniques. We have updated the wording in the abstract and introduction to better reflect this.

2. Different datasets containing electronic health records are quite different in the way they are structured/collected. For example, in the HES data there are many variables that should be non-missing (e.g. admission date etc) and, as indicated in this manuscript, the diagnosis and other codes should follow a predictable pattern of missing data. However, in other electronic health records (for example, UK data from general practices such as those held by the Clinical Practice Research Datalink) every GP encounter should be date stamped but most of the data recorded will relate to whatever information was recorded during that encounter (diagnoses, symptoms, treatments, tests, and so on). Some data items should result in others being recorded (e.g. a prescription should be accompanied by information about dose etc), in which case it would be possible to examine missing data in the way described; however, it is more difficult to see how useful the set visualisation tool would be for such datasets. I feel that this should be acknowledged somewhere in the manuscript – i.e. that although the tool may be extremely useful for some datasets, it may be less well suited to others as this depends very much on how the data are structured and collected.

The original structure of the Clinical Practice Research Datalink (CPRD) database differs from that of HES data. However, the CPRD scenario does have a good use case for set visualisation which we have now clarified further with a GP encounters example in the Discussion about using the tool to investigate “Missing data patterns in a relational database”. That example illustrates the steps a researcher should take and how set visualization would be useful (“For example, imagine a researcher’s database contained information about GP visits, diagnoses and prescriptions ...”).

Minor comments

1. As I understand it, this manuscript is partially aimed at epidemiologists / applied researchers using electronic health data for their research. Although entropy and information gain ratio are key concepts in information theory, they may not be widely known to epidemiological researchers. These terms are explained in this particular context but I wonder if it is worth giving a more general definition in the paper?

We have added definitions (see response to Reviewer 2).

2. Some acronyms are used but not defined (e.g. ESI)

Thank you for spotting that. We now spell “exclusive set intersections” in full and have deleted the ESI acronym.

Reviewer: 2

Dr. Danny Wu, University of Cincinnati

Comments to the Author:

The purpose of this paper was to demonstrate the effectiveness of ACE, a novel R and Java based interactive visualization tool, at identifying missing data from large scale electronic health records (EHRs) in the UK’s NHS digital. This tool included bar chart and heatmap visualization sets with data mining techniques to provide researchers with insights in missing patterns. The results showed that

ACE is effective in identifying widespread and rare data quality issues. The study can be utilized for both epidemiologists and hospital services to improve data quality and downstream analyses.

Overall, the paper is well written with the purpose and the relevance clearly identified. The implications and limitations of the present study are discussed in detail. While the case study and methodology need more specificity, the authors created comprehensive and relevant figures to allow the readers to have a greater visual understanding of the study design. My suggestions are listed below.

P4: A brief description of the methodology (e.g., visualization design choices and data mining techniques). is not given in the abstract.

We have added details of the visualizations to the Design. The data mining techniques (information gain ratio; entropy) are captured in the Results part of the Abstract.

P4 L3-5: This sentence could be better worded for clarity. Is there supposed to be a “that” after “understand”?

We have reworded that sentence.

P4 L9: Are the 16 million records freely available to UK researchers or any researchers?

The manuscript’s Data availability statement states “The dataset was provided by NHS Digital (request number DARS-NIC-17649-G0X4B-v0.6) and, due to data governance restrictions, cannot be made openly available.” Other researchers could obtain the same (or similar) data from NHS Digital if they make a suitable request and it is approved. NHS Digital makes a charge for such data.

P4 L14: Give an example of “unexpected gaps” in the study.

We have added that.

P4 L19: When stated “previously unknown to an Epidemiologist” should be stated as “previously unknown in the Epidemiology field”. Otherwise, it implies that one epidemiologist instigated the study and does not emphasize its relevance.

We have deleted the words “to an epidemiologist”. Our comment was specific to the lead epidemiologist for this particular data extract, and we do not wish to claim that this was unknown to the entire field.

P5 L8: Would the ACE be applicable to a different dataset? If so, what data manipulation effort would be needed?

This is covered in the Discussions, where we explain how ACE could be applied to longitudinal data and relational databases. That explanation has been expanded using the GP encounters example suggested by Reviewer 1.

P5 L22: Spell out TRIPOD when it is first used.

Done

P6 L4: What are the “core data fields” that basic counts only address?

We have added an example of a core field (NHS Number).

P6 L29: Provide examples of the existing tools to explain why their information and designs are not suitable to illustrate set intersections.

We have modified the wording to say “adopt designs that are unsuitable for showing intersections that involve many sets because Venn diagram or node-link representations are used”. Examples are provided by the references that we already provide.

P7 L22: When examining the data set, missing data may come from a smaller set of hospital practices. In other words, hospitals may have separate internal issues in data collection processes, leading to different prevalence of missing data. Consider addressing this in the discussion.

Thanks for this comment – we agree that is one of a variety of possible underlying causes of data being missing for one hospital and not others. We now mention three such possibilities in the Discussion (“and appropriate actions (e.g., fixing IT issues, modifying processes or staff training)”).

P9 L21: Explain at high level the information gain ratio and entropy calculations.

We have now done this (“IGR measures the change in entropy when a field is used to classify data”; “Entropy quantifies how cleanly data is divided into different classes”).

P14 L13: Consider adding another limitation in human-centered evaluation or formal usability testing, where multiple epidemiologists and other stakeholders are invited to perform tasks using the ACE to identify missing data patterns and provide feedback to the usability of the tool.

We have added that (see bullet #4).

VERSION 2 – REVIEW

REVIEWER	Cornish, Rosie University of Bristol, School of Social and Community Medicine
-----------------	--

REVIEW RETURNED	26-Oct-2022
GENERAL COMMENTS	The authors have addressed my previous comments and I have no further concerns.